# Energy Processes During *Rigor Mortis* in the Adductor Muscle of the Lion’s Paw Scallop (*Nodipecten subnodosus*): Effects of Seasonality and Storage Temperature

**DOI:** 10.3390/ani15202953

**Published:** 2025-10-12

**Authors:** Edgar Iván Jiménez-Ruiz, Víctor Manuel Ocaño-Higuera, María Teresa Sumaya-Martínez, Enrique Márquez-Ríos, Saúl Ruíz-Cruz, Dalila Fernanda Canizales-Rodríguez, Orlando Tortoledo-Ortiz, Alba Mery Garzón-García, José Rogelio Ramos-Enríquez, Santiago Valdez-Hurtado, María Irene Silvas-García, Nathaly Montoya-Camacho

**Affiliations:** 1Unidad de Tecnología de Alimentos, Secretaría de Investigación y Posgrado, Universidad Autónoma de Nayarit, Ciudad de la Cultura s/n, Tepic 63000, Mexico; edgar.jimenez@uan.edu.mx (E.I.J.-R.); teresa.sumaya@uan.edu.mx (M.T.S.-M.); 2Departamento de Ciencias Químico Biológicas, Universidad de Sonora, Blvd. Luis Encinas y Rosales s/n, Hermosillo 83000, Mexico; dalila.canizales@unison.mx (D.F.C.-R.); rogelio.ramos@unison.mx (J.R.R.-E.); nathaly.montoya@unison.mx (N.M.-C.); 3Departamento de Investigación y Posgrado en Alimentos, Universidad de Sonora, Blvd, Luis Encinas y Rosales s/n, Hermosillo 83000, Mexico; enrique.marquez@unison.mx (E.M.-R.); saul.ruizcruz@unison.mx (S.R.-C.); maria.silvas@unison.mx (M.I.S.-G.); 4Coordinación de Nutrición, Centro de Investigación en Alimentación y Desarrollo, A.C. Carretera Gustavo Astiazarán Rosas, No. 46, Hermosillo 83304, Mexico; otortoledo@ciad.mx; 5Ingeniería Industrial, Universidad del Valle, Sede Regional Caicedonia, Carrera 14 No 4-48, Caicedonia 762540, Valle del Cauca, Colombia; garzon.alba@correounivalle.edu.co; 6Unidad Académica Navojoa, Universidad Estatal de Sonora, Blvd, Manlio Fabio Beltrones 810, Col. Bugambilias, Navojoa 85875, Mexico; santiago.valdez@ues.mx

**Keywords:** *Nodipecten subnodosus*, energy metabolism, seasonality, storage temperature, transportation, *rigor mortis*

## Abstract

**Simple Summary:**

The lion’s paw scallop (*Nodipecten subnodosus*) is a high-value mollusk species cultivated in northwestern Mexico, but its postmortem biochemical changes during storage are still poorly understood. This study investigated the effect of seasonality, storage temperature, and transportation on energy-related metabolites associated with rigor mortis in the adductor muscle. Scallops were stored at 0, 5, and 10 °C for 48 h across all four seasons, and the concentrations of ATP, ADP, AMP, glycogen, arginine phosphate, and arginine were quantified. The adenylate energy charge (AEC) was also determined in freshly harvested and transported scallops. The results showed that ATP depletion and AEC decline were strongly affected by storage temperature and seasonality, with significant implications for the postharvest quality and shelf life of this species.

**Abstract:**

The lion’s paw scallop (*Nodipecten subnodosus*) is a commercially valuable pectinid whose postharvest quality strongly depends on storage and handling conditions. This study investigated the combined effects of seasonality, postmortem time, and storage temperature on energy metabolism in the adductor muscle, focusing on metabolites associated with *rigor mortis* and freshness. Adult scallops (~10 cm shell height) were harvested in four seasons (spring, summer, autumn, winter), transported under commercial conditions for approximately 2 h, and stored at 0, 5, and 10 °C for 48 h. Muscle samples were collected every 8 h and analyzed for ATP, ADP, AMP, glycogen, arginine phosphate (Arg-P), and free arginine using HPLC and enzymatic assays. In addition, the adenylate energy charge (AEC) was determined in freshly harvested and post-transport specimens. Initial ATP concentrations ranged from 4.2 to 6.5 µmol/g, with higher levels in winter, while Arg-P varied from 3.1 to 4.8 µmol/g. Seasonality significantly influenced all metabolites except arginine, and transport markedly reduced ATP and AEC, particularly in spring and autumn. Storage at 0 °C resulted in rapid ATP depletion (<1.0 µmol/g within 12 h) and AMP accumulation (>3.0 µmol/g), indicating accelerated energy collapse. In contrast, scallops stored at 5 and 10 °C maintained ATP levels above 2.5 µmol/g for up to 24 h, delaying *rigor mortis*, reducing postmortem contraction, and preserving muscle texture and appearance. Overall, these findings demonstrate that moderate refrigeration represents a physiologically suitable and technologically advantageous strategy to optimize scallop postharvest handling, extend shelf life, and enhance product quality for the fresh seafood market.

## 1. Introduction

The lion’s paw scallop, *Nodipecten subnodosus* (Sowerby I, 1835), is one of the largest pectinid species, reaching up to 218 mm in shell height (SH) and inhabiting tropical and subtropical waters along the Pacific and Gulf of California coasts of Baja California, Mexico, down to Peru [1,2,3]. In aquaculture conditions, it attains a mean SH of 140 mm in approximately three years. This high-value species represents a key target for aquaculture development in the Baja California Peninsula [1,4]. Its commercial appeal stems from its large adductor muscle, rapid growth (reaching commercial size of 7 cm within 8 months), and premium market price (approximately USD 25/kg), as well as its favorable sensory attributes [5,6,7]. These characteristics cause stimulated increasing efforts to optimize its cultivation, processing, and postharvest management, particularly as production shifts from fishery-based exploitation to aquaculture-based supply chains.

Postmortem muscle quality in aquatic species is strongly influenced by physiological status at harvest, environmental stressors, and storage conditions [8]. The death of the organism triggers a cascade of postmortem biochemical changes, which include the cessation of circulation and oxygen supply, a shift to anaerobic metabolism, rapid ATP depletion, accumulation of lactic acid, pH decline, and the initiation of *rigor mortis*. These changes ultimately affect the physicochemical integrity of the muscle, determining its texture, water-holding capacity, and shelf-life [9,10,11]. Among these processes, *rigor mortis* plays a central role. It is defined as the progressive stiffening of the muscle caused by the irreversible formation of actomyosin complexes following ATP depletion, resulting in the loss of elasticity and contractile function [12,13,14]. The onset, intensity, and duration of *rigor mortis* are governed by species-specific metabolism, physiological condition at slaughter, and postharvest handling, particularly temperature [15,16].

In pectinids, energy metabolism after death relies primarily on two biochemical reserves: arginine phosphate, a high-energy phosphagen unique to invertebrates, and glycogen. Arginine phosphate supports rapid ATP regeneration via arginine kinase during the early postmortem phase, while glycogen fuels longer-term ATP synthesis through anaerobic glycolysis [16,17,18,19,20]. The mobilization of these substrates governs the postmortem energy balance and, consequently, the progression of *rigor mortis*. Seasonal fluctuations in environmental temperature and reproductive stage have been shown to influence pre-harvest energetic reserves and stress tolerance in mollusks [21,22], potentially affecting postmortem biochemical responses.

Previous research on postmortem energy metabolism has focused on other molluscan species such as *Zygochlamys patagonica* [23], *Mizuhopecten yessoensis* [24], *Haliotis midae* [25], *Haliotis discus* [26], *Bathymodiolus platifrons* [27], *Patinopecten yessoensis*, and *Pecten albicans* [20], demonstrating species-specific patterns in ATP degradation, arginine phosphate utilization, and rigor onset under different storage conditions. However, little is known about how these processes are modulated in *Nodipecten subnodosus*, particularly under the combined effects of seasonal variation and refrigerated storage.

Understanding the postmortem energetic profile of *Nodipecten subnodosus* is essential for optimizing storage protocols, minimizing quality loss, and improving its marketability in the fresh seafood sector. Therefore, the aim of this study was to evaluate the effects of seasonality and storage temperature on the concentrations of key energy-related metabolites and the development of *rigor mortis* in the adductor muscle of *N. subnodosus*.

## 2. Materials and Methods

### 2.1. Experimental Organisms

Adult specimens of *Nodipecten subnodosus* (approximately 10 cm in shell height) were obtained from a suspended lantern net culture system located in Bahía Tortugas, Baja California Sur (BCS), Mexico (27°41′30″ N, 114°53′45″ W). Specimen collection was carried out with the assistance of trained aquaculture personnel, following the harvesting technique normally used at the site.

Sampling was conducted in four separate seasons: spring, summer, autumn, and winter. For each sampling, 120 individuals were collected. Six specimens were immediately frozen in liquid nitrogen to represent the initial condition (harvested organisms), while the remaining scallops were transported to a processing facility equipped for adductor muscle extraction (Marimex del Pacífico, S.A. de C.V., La Paz, BCS). The transport was performed by specialized personnel and lasted approximately 2 h. During this time, the scallops were kept in plastic containers covered with moistened cotton cloth to minimize desiccation. Ambient temperature during transport varied by season, with average values of 15 °C in spring, 19 °C in summer, 22 °C in autumn, and 18 °C in winter. This additional information provides a clearer context for interpreting the post-transport (0 h) energetic condition of the organisms.

### 2.2. Storage Experiment

Following shucking, the adductor muscles were immediately packed in polyethylene bags, with six muscles per bag. For each season, samples were randomly assigned to three storage temperatures: 0, 5, and 10 °C. The storage experiment was conducted over a 48 h period, during which samples were collected every 8 h for biochemical analysis, since preliminary data and previous studies [28,29,30] have shown that the most relevant postmortem biochemical changes in scallops, including those related to *rigor mortis* energetics, occur within this time frame. An exception was made for the determination of adenylate energy charge (AEC) which were evaluated only in initial (harvested) and post-transport (0 h) in each season.

At each sampling point, adductor muscle samples were flash-frozen in liquid nitrogen and stored at −80 °C until analysis of ATP, ADP, AMP, AEC, glycogen, arginine phosphate, and arginine concentrations. The 0 h time point corresponded to samples taken immediately after transport and muscle extraction. For storage at 0 °C, samples were placed directly on crushed ice. For 5 and 10 °C conditions, calibrated General Electric refrigerators (Model GMR02BANMWH, Singapore) were used to maintain the desired temperature.

### 2.3. ATP, ADP, AMP

Extraction, identification, and quantification of adenosine 5′-triphosphate (ATP), adenosine 5′-diphosphate (ADP), and adenosine 5′-monophosphate (AMP) were performed using the reverse-phase high-performance liquid chromatography (HPLC) method described by Ryder [31], with modifications. Muscle samples were homogenized in 0.6 M perchloric acid, and the resulting extracts were filtered and diluted prior to injection.

A volume of 20 µL of the diluted extract was injected into a Varian ProStar 240 chromatograph (Varian Inc., Lake Forest, CA, USA) equipped with a C18 reversed-phase column (4.6 × 150 mm; Varian Inc., CA, USA). The mobile phase consisted of a phosphate buffer (0.04 M KH_2_PO_4_ and 0.06 M K_2_HPO_4_), with a flow rate of 1.0 mL/min. Detection was carried out at 254 nm using a UV-Vis detector (Varian ProStar 325, Varian Inc., CA, USA). Retention times were compared against commercially available standards to identify each nucleotide.

### 2.4. AEC

The adenylate energy charge (AEC) was calculated to evaluate the energetic status of the adductor muscle during postmortem storage. AEC was determined using the concentrations of ATP, ADP, and AMP obtained by HPLC, following the formula proposed by Guida et al. [32]:AEC = [(ATP) + ½ (ADP)]/[(ATP) + (ADP) + (AMP)]
where ATP = adenosine 5′-triphosphate, ADP = adenosine 5′-diphosphate, and AMP = adenosine 5′-monophosphate. AEC values were calculated for both freshly harvested (initial) and post-transport samples at defined storage times (0, 8, 24, and 48 h) across all seasonal replicates.

### 2.5. Glycogen

Glycogen content in the adductor muscle was quantified following the method of Racotta et al. [33], with minor modifications. Briefly, 0.3 g of muscle tissue was homogenized in 1 mL of cold 10% trichloroacetic acid (TCA) using an Ultra-Turrax T18 Basic homogenizer (IKA Works Inc., Wilmington, NC, USA) for 1 min. The homogenate was centrifuged at 3000× *g* for 15 min at −5 °C (IEC-MULTI RF, Thermo Fisher Scientific, Asheville, NC, USA). A 0.1 mL aliquot of the supernatant was mixed with 1 mL of 95% ethanol and centrifuged again under the same conditions. The resulting pellet was dried in a vacuum concentrator (Vacufuge Plus, Eppendorf, Hamburg, Germany) at 45 °C for 15 min, and then resuspended in 0.1 mL of distilled water.

For colorimetric quantification, 1 mL of anthrone reagent (0.1% in 76% sulfuric acid) was added, and the mixture was incubated at 90 °C for 5 min. The reaction was stopped by cooling the tubes in ice. Absorbance was measured at 620 nm using a UV-Visible spectrophotometer (Cary 100 Bio, Varian Inc., Mulgrave, VIC, Australia). Glycogen concentration was expressed as glycosyl units, using a glucose standard curve.

### 2.6. Arginine Phosphate

Arginine phosphate content was determined using the same perchloric acid extracts prepared for nucleotide analysis. Quantification was performed by high-performance liquid chromatography (HPLC) according to the method of Viant et al. [34]. Analyses were carried out using a Varian ProStar 240 chromatograph (Varian Inc., Lake Forest, CA, USA) equipped with a SUPELCOSIL^®^ LC-NH_2_ column (4.6 × 250 mm; Supelco, Bellefonte, PA, USA). The mobile phase consisted of a 0.02 M KH_2_PO_4_ buffer (pH 2.6) and acetonitrile (72:28, *v*/*v*), delivered at a flow rate of 1.2 mL/min. Detection was performed at 205 nm using a UV-Vis detector (Varian ProStar 325, Varian Inc., Lake Forest, CA, USA). Identification and quantification of arginine phosphate were based on comparison with analytical-grade standards.

### 2.7. Arginine

Arginine concentration was quantified using the same perchloric acid extracts employed for nucleotide analysis, following the method of Sato et al. [35]. A 20 µL aliquot of each extract was injected into a Varian HPLC system (Varian Inc., Lake Forest, CA, USA) equipped with a reversed-phase C18-A column (4.6 × 100 mm; Varian Inc.) maintained at 40 °C using a Metatherm^®^ column heater. A gradient elution was performed using two mobile phases: solution A (20% acetonitrile in 0.25 M Tris-HCl buffer, pH 9.5) and solution B (80:20 acetonitrile:water, *v*/*v*). The linear gradient progressed from 0% to 50% solution B over 20 min, followed by a final washing step with 100% solution B. Detection was carried out by fluorescence, using a Varian ProStar 363 detector with excitation and emission wavelengths of 325 and 425 nm, respectively. Identification and quantification were based on the retention time and peak area of analytical-grade arginine standards.

### 2.8. Statistical Analysis

The experiment was structured under a completely randomized design with a factorial arrangement to evaluate the effects of season (spring, summer, autumn, and winter), storage temperature (0, 5, and 10 °C), and postmortem storage time (0, 8, 16, 24, 32, 40, and 48 h) on the concentrations of energy-related metabolites and the progression of *rigor mortis* in *Nodipecten subnodosus*. For each treatment combination, six biological replicates were analyzed (n = 6). The data were subjected to analysis of variance (ANOVA) to assess the main effects and interactions among factors. Differences were considered statistically significant at *p* < 0.05. When significant differences were detected, Tukey’s multiple range test was applied to perform pairwise comparisons among means. All statistical analyses were carried out using the NCSS 2000 software package [36].

## 3. Results and Discussions

### 3.1. ATP, ADP and AMP

Adenosine triphosphate (ATP) is the primary energy carrier in muscle tissue and a key determinant of postmortem biochemical stability. After death, the progressive hydrolysis of ATP initiates *rigor mortis* and affects various quality attributes, including texture, water-holding capacity, and shelf-life [10]. In mollusks, ATP degradation also serves as a biochemical marker of physiological status and stress response [37,38]. Recent studies on scallops such as *Pecten maximus* and *Argopecten irradians* have demonstrated that postmortem ATP stability is influenced by storage temperature, reproductive stage, and seasonal variation [39,40].

In this study, ATP concentrations in the adductor muscle varied significantly (*p* < 0.05) with both storage temperature and season (Figure 1). Initial ATP levels (from lagoon-harvested organisms) were highest in summer and lowest in spring. A significant decline in ATP was observed after transport to the shucking site (day 0) in spring, summer, and autumn (*p* < 0.05), while winter samples showed no significant changes (*p* > 0.05). During storage, ATP levels decreased in all treatments but remained significantly higher at 5 and 10 °C compared to 0 °C (*p* < 0.05). A transient increase in ATP was detected at 8 h in spring and autumn samples stored at 5 and 10 °C, whereas at 0 °C the decline was continuous and more pronounced.

The initial drop in ATP during transport reflects the metabolic cost of handling stress and seasonal physiological conditions. In spring, this decline may be linked to cold-induced thermal stress, as lagoon temperatures (15 °C) fall below the species’ thermal optimum of 22 °C [41,42]. In summer and autumn, reduced ATP levels may be attributed to gametogenic activity and recent spawning, which increase energy demand and deplete reserves [19,21].

The postmortem ATP increase observed at 8 h under moderate temperatures is consistent with the activity of the arginine kinase system, which regenerates ATP from ADP using phosphoarginine. This transient buffering effect has been previously reported in *Zygochlamys patagonica* and *Pecten maximus* stored at similar temperatures [23,39]. In contrast, storage at 0 °C may suppress enzymatic activity, limiting ATP regeneration and accelerating degradation.

Previous studies have emphasized the importance of ATP levels in early postmortem stages, as they influence the progression of other biochemical changes [43,44,45]. It is widely accepted that *rigor mortis* begins when ATP concentrations fall below 1.0 µmol/g [10,11]. In the present study, ATP levels remained above this threshold for at least 32 h at 5 and 10 °C in summer, autumn, and winter samples, suggesting a delayed onset of *rigor mortis* under these conditions. These findings support previous reports indicating that moderate chilling helps preserve biochemical integrity by maintaining ATP levels longer than ice storage [37]. Therefore, short-term storage at 5 °C may be strategically advantageous in postharvest management to extend the freshness of *Nodipecten subnodosus*.

Adenosine diphosphate (ADP) is a key intermediate in postmortem energy metabolism, functioning both as a degradation product of ATP and a precursor for its resynthesis via the phosphagen system [46]. In marine invertebrates, ADP levels fluctuate dynamically during early postmortem phases and are closely linked to the tissue’s capacity for transient energy buffering. Monitoring ADP dynamics thus provides insight into the energy regeneration potential and overall metabolic state of muscle during cold storage [47]. Studies in scallops and abalone suggest that ADP patterns are sensitive to both extrinsic factors such as temperature and intrinsic factors like reproductive stage and season [39,40].

In this study, ADP concentrations varied significantly according to season and storage temperature (Figure 2). The highest initial levels were recorded in spring and autumn, coinciding with the lowest ATP levels (*p* < 0.05). A significant increase in ADP was observed during transport in spring (*p* < 0.05), while levels remained relatively stable in the other seasons (*p* > 0.05). During storage, ADP concentrations declined at 8 h in spring and autumn samples kept at 5 and 10 °C (*p* < 0.05), in parallel with the observed ATP rebound. At 0 °C, ADP levels either remained elevated or increased further over time, with no transient decrease.

These patterns reflect a temperature-dependent balance between ATP degradation and regeneration pathways. The postmortem decline of ADP at 8 h suggests its conversion back to ATP via arginine kinase, supported by concurrent ATP increases and stable or reduced AMP concentrations. This mechanism is consistent with findings in *Pecten maximus* and *Crassostrea gigas*, where ADP turnover at moderate temperatures indicates active phosphagen buffering [48,49]. Conversely, the persistence of high ADP levels at 0 °C indicates enzyme inhibition and limited ATP regeneration capacity.

The elevated initial ADP levels in spring and autumn may also relate to seasonal reproductive physiology [50]. These periods coincide with gonadal development and spawning in *Nodipecten subnodosus*, which impose high metabolic demands and reduce somatic energy reserves [21,22]. As a result, a greater proportion of the muscle adenylate pool may exist in the ADP form at harvest, predisposing tissues to faster energy depletion during storage.

Altogether, ADP dynamics highlight the benefits of moderate refrigeration (5–10 °C) in supporting early postmortem energy balance. ADP turnover at these temperatures indicates residual enzymatic activity that may delay *rigor mortis* onset and preserve muscle functionality [51,52]. In contrast, storage at 0 °C may prematurely arrest energy metabolism, leading to rapid muscle stiffening and reduced shelf-life.

Adenosine monophosphate (AMP) is the final degradation product in the adenylate energy pathway and serves as a biochemical indicator of irreversible energy loss [37]. Unlike ATP and ADP, AMP cannot be directly converted back to ATP through the phosphagen system and thus accumulates progressively during postmortem storage. Its concentration reflects the collapse of cellular energy homeostasis and correlates with the onset of *rigor mortis* [10,51]. Elevated AMP levels are associated with reduced product quality, including tougher texture and shortened shelf-life in mollusks.

In the present study, AMP concentrations increased significantly in all treatments throughout storage, with the most pronounced accumulation occurring at 0 °C (Figure 3). Initial AMP levels were highest in autumn (*p* < 0.05), and transport further increased AMP in spring and autumn samples (*p* < 0.05), suggesting that seasonal physiological stress exacerbates ATP degradation. At 8 h of storage, a sharp rise in AMP was observed at 0 °C across all seasons (*p* < 0.05), coinciding with the lowest ATP levels. In contrast, AMP accumulation was slower and less severe at 5 and 10 °C, particularly in winter samples.

These findings support the hypothesis that enzyme inhibition at low temperatures limits ATP regeneration, leading to continued ATP depletion and AMP buildup. The absence of ATP rebound and continuous AMP accumulation at 0 °C contrasts with the transient buffering observed at 5 and 10 °C, where arginine kinase remains more active. Similar trends have been reported in *Pecten maximus* and *Crassostrea gigas*, where AMP buildup serves as a reliable marker of irreversible energy collapse and rigor development during cold storage [48,53].

High AMP levels in spring and autumn may also reflect reproductive investment, as both seasons correspond to advanced gametogenic activity or recent spawning events in *Nodipecten subnodosus*. These physiological states deplete muscle energy reserves, increase stress sensitivity, and accelerate postmortem metabolic deterioration [21,22]. Overall, AMP profiles confirm that while low-temperature storage (0 °C) may preserve microbial safety, it compromises biochemical integrity. The sustained rise in AMP at 0 °C indicates rapid freshness loss and a limited quality window. Conversely, the slower AMP accumulation at 5 and 10 °C suggests that moderate refrigeration can delay postmortem degradation and extend product shelf-life in *N. subnodosus*.

It is important to highlight that the postmortem changes observed in *Nodipecten subnodosus* are consistent with those reported for other scallop species. In *Pecten maximus*, rapid ATP degradation during chilled storage has been described, with a parallel decline in adenylate energy charge (AEC), confirming the sensitivity of this parameter as an indicator of freshness and metabolic status [30]. Similarly, in *Placopecten magellanicus*, a sharp decrease in ATP and the accumulation of AMP postmortem have been documented, reinforcing the conserved role of adenylate catabolism across scallops [54]. These observations support our findings that low storage temperatures accelerate the irreversible depletion of energy metabolites, whereas moderate refrigeration better preserves nucleotide balance. The faster depletion of ATP observed in *N. subnodosus* during summer and autumn may be attributed to species-specific physiological stress associated with reproductive activity.

### 3.2. Adenylate Energy Charge (AEC)

The AEC, first proposed by Atkinson [55], is a complementary indicator used to assess capture-induced stress and to provide insights into cellular stress responses. The AEC reflects the metabolic energy status of a cell and is calculated from the relative concentrations of ATP, ADP, and AMP, yielding a value between 0 and 1 [32]. Fleury et al. [56] reported AEC values ranging from 0.8 to 1.0 in healthy organisms, 0.5 to 0.7 in moderately stressed individuals, and below 0.5 in severely stressed organisms. While moderately stressed organisms may exhibit reduced growth and reproductive capacity, they generally retain the potential for recovery. In contrast, severely stressed individuals are unable to grow or reproduce, and their viability may be compromised, even after being returned to normal, stress-free conditions. In mollusks, the AEC is particularly sensitive to handling stress, reproductive stage, and storage conditions, with values above 0.7 typically associated with high muscle quality [57].

The AEC values for organisms sampled before and after transport across the spring, summer, autumn, and winter seasons are presented in Table 1. In the present study, AEC values varied significantly with both season and transport conditions (*p* < 0.05). The lowest AEC values were observed in spring and autumn specimens collected directly from the lagoon (*p* < 0.05), likely reflecting the influence of reproductive effort and thermal stress; however, these organisms retained the capacity for recovery. In contrast, organisms collected during summer and winter exhibited high AEC values, indicating a robust energetic condition consistent with healthy physiological status. A study by Ocaño-Higuera et al. [58] on *Nodipecten subnodosus* collected at the end of summer (September) reported an AEC value of 0.8, aligning with the results obtained for this season in the present study. Moreover, a significant decline in AEC was observed after transport during spring, summer, and autumn (*p* < 0.05), which coincided with a marked decrease in ATP levels (*p* < 0.05). In contrast, winter samples maintained stable AEC values. A previous study conducted on *Crassostrea gigas* also demonstrated a seasonal effect on AEC in organisms subjected to emersion (i.e., short-term desiccation stress). The most energetically affected individuals were collected in spring (May) and summer (July), while those harvested in winter (January) retained optimal energetic conditions [59]. Similar effects of desiccation-induced stress during transport were reported by Maguire et al. [60] and Ocaño et al. [58] in *Pecten maximus* and *N. subnodosus*, respectively, collected during spring and summer.

These seasonal differences are consistent with the reproductive biology of *Nodipecten subnodosus*, as scallops invest substantial energy in gametogenesis and spawning during spring and autumn. This energetic investment reduces somatic energy reserves, resulting in lower initial AEC values and diminished postmortem recovery capacity [21,22]. The combination of physiological factors and handling stress during transport may thus compromise freshness even before cold storage begins.

### 3.3. Glycogen

Glycogen is the primary carbohydrate reserve in molluscan muscle and plays a critical role in sustaining ATP production postmortem, particularly via anaerobic glycolysis when oxygen is no longer available [61,62]. Its depletion rate influences not only ATP regeneration but also pH stability and lactic acid accumulation, with direct implications for the onset of *rigor mortis* and overall muscle quality [63,64]. The baseline glycogen content and its postharvest utilization are strongly influenced by the organism’s physiological status, environmental temperature, and reproductive stage [65,66,67].

Figure 4 shows the glycogen concentrations in the adductor muscle of *Nodipecten subnodosus* across different storage temperatures and seasons. The highest initial levels were observed in spring and summer, with concentrations of 6.43 ± 1.43% and 7.69 ± 0.58%, respectively. These values were not statistically different (*p* > 0.05), but significantly higher than those in autumn (2.05 ± 0.14%) and winter (2.07 ± 0.21%) (*p* < 0.05). Beltrán-Lugo et al. [22] reported similar seasonal variations, with values of 3.51%, 7.87%, 2.21%, and 0.79% for spring (April), summer (September), autumn (November), and winter (January), respectively. Differences may be attributed to the specific month of sampling or regional aquaculture conditions.

Transport of the organisms did not significantly affect glycogen levels (*p* > 0.05); however, a decreasing trend was observed at 5 °C and 10 °C during the early storage stages (*p* < 0.05). At 0 °C, glycogen concentrations remained relatively unchanged throughout the storage period (*p* > 0.05). This trend was consistent across all seasons and mirrored ATP profiles—higher ATP preservation at lower temperatures corresponded with greater glycogen degradation, supporting its role in postmortem energy metabolism [10]. These results highlight the importance of glycogen as a postmortem energy substrate in scallops, which are known to store appreciable amounts of this compound in the adductor muscle [19,33]. In addition, glycogen content has been used, similar to the adenylate energy charge (AEC), as a biomarker of chronic stress in pectinids, such as seasonal physiological shifts [58,60]. For instance, Ansell [68] reported that *Pecten maximus* glycogen levels varied with both location and season, reaching a maximum (~5.2%) in late summer (september) and a minimum (~0.55%) in late winter (March), consistent with the seasonal fluctuations observed in this study.

It is important to highlight that comparable postmortem nucleotide degradation patterns have been described in the Patagonian scallop *Zygochlamys patagonica*. Massa et al. [28] reported that ATP levels in the adductor muscle decline progressively during cold storage, accompanied by AMP accumulation and a corresponding reduction in AEC. These trends are consistent with the seasonal profiles observed in *Nodipecten subnodosus*, although the rate of ATP depletion in *Z. patagonica* was generally slower. Such differences may reflect species-specific physiology and adaptation to their natural environments, emphasizing that while adenylate catabolism is a conserved mechanism among scallops, the kinetics of energetic collapse are modulated by intrinsic biological factors and harvest conditions.

Also, it is also important to note that similar trends have been documented in the Pacific oyster *Crassostrea gigas*. Yokoyama et al. [69] demonstrated that ATP depletion in oyster tissues is strongly influenced by storage temperature, with lower temperatures accelerating nucleotide breakdown and leading to faster declines in AEC. This finding parallels our observations in *Nodipecten subnodosus*, where ice storage at 0 °C resulted in a more rapid loss of energy charge compared with moderate refrigeration. However, the initial AEC values in *C. gigas* were lower than those recorded in *N. subnodosus*, which may reflect interspecific differences in energy reserves and metabolic strategies. Together, these comparisons reinforce the usefulness of AEC as a robust indicator of freshness across mollusks, while underscoring species-specific variability in the rate of postmortem energy depletion.

### 3.4. Arginine Phosphate

Arginine phosphate (Arg-P) serves as the principal phosphagen in mollusks, functioning analogously to phosphocreatine in vertebrates. It plays a crucial role in sustaining ATP levels during the early postmortem phase, when mitochondrial oxidative phosphorylation ceases [70]. Through the enzymatic action of arginine kinase, Arg-P transfers a phosphate group to ADP, rapidly regenerating ATP and producing free arginine as a byproduct [17,71]. This phosphagen system is particularly important in scallop species, which undergo intense muscular activity during harvesting and handling. Therefore, the rate of Arg-P depletion postmortem may serve as a biochemical indicator of residual energy capacity and the onset of metabolic collapse [51].

Figure 5 illustrates Arg-P concentrations in the adductor muscle of *Nodipecten subnodosus* collected across four seasons and stored at different temperatures. Initial values ranged from 1.65 µmol/g in spring to 8.17 µmol/g in summer. These levels are consistent with those reported for *Patinopecten yessoensis* (~5 µmol/g) by Kawashima and Yamanaka [30], but substantially lower than the ~17.5 µmol/g and ~14 µmol/g observed in *Pecten albicans* and *P. yessoensis*, respectively, by Wongso et al. [20]. Additionally, values close to 7 µmol/g were reported for *Chlamys nobilis* in spring (April), which aligns with the present study’s findings in winter (8.17 µmol/g) and summer (5.86 µmol/g) samples.

These interspecific and seasonal differences likely stem from physiological and genetic variation, as well as environmental influences such as seawater temperature, reproductive cycle, and food availability [72]. For instance, elevated Arg-P levels in summer may reflect enhanced energy storage during gonadal development, while lower concentrations in spring could indicate post-spawning depletion. These seasonal metabolic patterns have been consistently reported in scallops and other bivalves [41,73,74,75], reinforcing the importance of sampling context in postmortem biochemical interpretation [5].

Transport to the shucking facility did not significantly affect Arg-P levels in any season (*p* > 0.05). However, a sharp and statistically significant decrease in Arg-P was observed after 8 h of refrigerated storage at all temperatures and in all seasonal groups (*p* < 0.05). This behavior is consistent with the rapid utilization of Arg-P to regenerate ATP during the initial postmortem period, facilitated by arginine kinase activity [23,26,76].

Interestingly, a transient increase in ATP concentrations was detected at 8 h (particularly in spring and autumn samples), which may be partially explained by the hydrolysis of Arg-P. Following this peak, Arg-P levels in spring samples declined sharply to 0.09–0.40 µmol/g across all storage conditions, indicating near-total depletion. In contrast, summer, autumn, and winter samples retained higher Arg-P concentrations over time (0.36–1.92 µmol/g), suggesting greater metabolic stability and energy buffering capacity.

In autumn samples, storage at 5 °C and 10 °C initially resulted in significantly lower Arg-P concentrations compared to 0 °C (*p* < 0.05); however, by the end of the storage period, no significant differences among temperatures were detected (*p* > 0.05), indicating convergence toward a common energetic baseline. This trend is consistent with previous findings in *Patinopecten yessoensis* and *Pecten maximus*, where Arg-P depletion occurred rapidly during the early *postmortem* window, followed by stabilization as phosphagen reserves were exhausted [77,78,79].

It is important to highlight that the transient ATP rebound observed in *N. subnodosus* at moderate storage temperatures was closely associated with the mobilization of arginine phosphate (Arg-P), a finding consistent with previous studies in other mollusks. In scallops and abalone, Arg-P has been shown to act as a phosphagen buffer, regenerating ATP through the arginine kinase reaction during the early postmortem phase [30]. These reports, in agreement with our results, emphasize the critical role of the phosphagen system in stabilizing the adenylate pool immediately after harvest. However, the sharper decline of Arg-P observed in *N. subnodosus* during summer and autumn suggests a limited capacity to sustain energy homeostasis under physiologically stressful conditions, such as reproduction, which accelerates the collapse of ATP and AEC.

From a technological perspective, characterizing Arg-P dynamics is essential for understanding the biochemical mechanisms that influence early postmortem muscle quality. Rapid Arg-P depletion may accelerate pH decline, *rigor mortis* onset, and structural protein denaturation. Conversely, species or batches with higher initial Arg-P levels may benefit from enhanced short-term energy buffering, contributing to better preservation outcomes during chilled storage. These findings highlight the relevance of Arg-P as a metabolic quality indicator and a potential target for postharvest handling optimization in scallop supply chains.

### 3.5. Arginine

Free arginine plays a pivotal role in molluscan postmortem metabolism, acting as the substrate for arginine kinase (the key enzyme responsible for phosphagen) mediated ATP regeneration [72]. During the early postmortem period, arginine is liberated from the hydrolysis of arginine phosphate (Arg-P), and its concentration dynamics reflect the tissue’s energy turnover and its capacity to buffer ATP depletion [26,70]. Beyond its postmortem relevance, arginine also contributes to nitrogen metabolism and osmoregulation in living mollusks; however, its levels after death are primarily dictated by phosphagen mobilization and subsequent degradation [80,81].

Figure 6 presents the free arginine concentrations in *Nodipecten subnodosus* during storage at 0, 5, and 10 °C across four seasons. Initial concentrations in freshly harvested organisms from the lagoon were statistically similar across all seasons (*p* > 0.05), averaging approximately 10 µmol/g. These values are consistent with those reported by Kawashima and Yamanaka [30] for *Patinopecten yessoensis*, but lower than those observed by Watanabe et al. [26] in *Haliotis discus* (~22 µmol/g) and by Wongso et al. [20] in *Chlamys nobilis*, *Pecten albicans*, and *P. yessoensis* (~24–45 µmol/g). These interspecific differences may arise from variations in baseline Arg-P reserves, species-specific metabolic rates, or physiological conditions prior to harvest. Notably, in the present study, transportation from the lagoon to the shucking facility did not significantly affect arginine levels (*p* > 0.05), suggesting that this step did not impose substantial metabolic stress.

During storage, all seasonal groups exhibited a similar pattern in arginine dynamics. A significant increase in arginine concentration was detected at 8 h of storage at both 5 °C and 10 °C (*p* < 0.05), consistent with the enzymatic action of arginine kinase, which releases free arginine as it regenerates ATP from ADP [23,26]. This trend parallels the observed decrease in Arg-P at the same time point, further supporting the link between Arg-P hydrolysis and transient ATP maintenance. After this peak, arginine concentrations either stabilized or gradually declined, depending on the season and storage temperature.

The magnitude and persistence of arginine accumulation were strongly modulated by storage temperature. At 10 °C, higher concentrations were detected, reflecting accelerated Arg-P hydrolysis and greater metabolic turnover, consistent with residual arginine kinase and glycolytic activity at moderate refrigeration [30]. In contrast, storage at 0 °C limited arginine release but markedly accelerated ATP degradation, resulting in faster AMP and IMP accumulation compared with 5 and 10 °C [69]. These findings suggest that extremely low temperatures suppress the enzymatic pathways necessary for balanced nucleotide turnover, thereby precipitating an earlier energetic collapse. From a technological standpoint, this moderated arginine response and accelerated ATP breakdown at 0 °C may be advantageous by slowing glycolysis and pH decline, ultimately delaying the onset of *rigor mortis*. 

From a practical standpoint, the present findings have direct relevance for postharvest handling of *Nodipecten subnodosus*. The observation that moderate refrigeration (5–10 °C) preserves ATP levels and delays the onset of *rigor mortis* more effectively than ice storage at 0 °C suggests that industry protocols could be optimized by adopting storage conditions that avoid extreme chilling. Such an approach may contribute to improved sensory quality, reduced textural deterioration, and extended shelf-life of scallop adductor muscles, while still ensuring microbiological safety. Furthermore, the integration of adenylate and phosphagen profiling as objective freshness indicators provides valuable tools for quality assurance systems, enabling more precise monitoring of biochemical status during the cold chain. These practical implications underscore the importance of tailoring postharvest strategies to the physiological characteristics of the species and seasonal context, thereby enhancing product value and competitiveness in scallop supply chains.

## 4. Conclusions

By integrating adenylate and phosphagen profiling (ATP, ADP, AMP, Arg-P, free arginine, glycogen, and AEC), this study shows that storage temperature and season jointly determine postmortem energetic homeostasis in the adductor muscle of *Nodipecten subnodosus*. Moderate refrigeration (5–10 °C) consistently preserves ATP and delays rigor onset relative to 0 °C, a condition under which suppressed enzymatic activity limits ATP regeneration and accelerates irreversible energy loss. Seasonal physiology explains lower initial reserves and greater handling sensitivity in spring and autumn. The transient ATP rebound at ~8 h observed at 5–10 °C coincides with Arg-P hydrolysis and increased free arginine, whereas AMP accumulates fastest at 0 °C, marking energetic collapse. Interpreted alongside ATP and Arg-P, free arginine emerges as a sensitive early biomarker of postmortem status. Collectively, these metabolites provide an operational framework for monitoring freshness and guiding postharvest decisions; practically, we recommend moderate refrigeration during the first 24–32 h post-harvest, with protocols tailored to seasonal physiological state. Future work should couple these biochemical indicators with textural, sensory, and microbiological endpoints to refine freshness metrics and extend shelf-life under commercial conditions.

## Figures and Tables

**Figure 1 animals-15-02953-f001:**
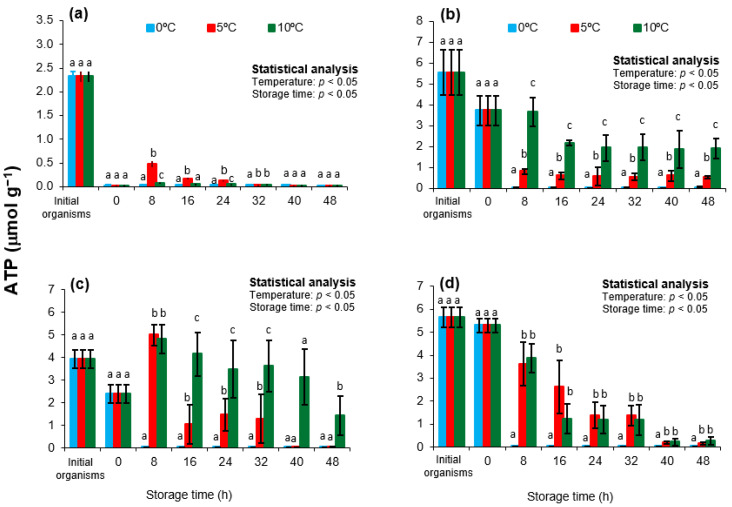
Adenosine triphosphate (ATP) concentrations in the adductor muscle of the lion’s paw scallop (*Nodipecten subnodosus*) harvested in different seasons and stored at three temperatures (0 °C, 5 °C, and 10 °C) for 48 h. Seasonal profiles are presented for (**a**) spring, (**b**) summer, (**c**) autumn, and (**d**) winter. Values are expressed as mean ± standard deviation (n = 6). “Initial” denotes specimens sampled directly from the lagoon at harvest, whereas “0 h” refers to samples collected immediately after transport to the shucking facility. Different lowercase letters above the bars indicate significant differences (*p* < 0.05) among storage temperatures at the same sampling time.

**Figure 2 animals-15-02953-f002:**
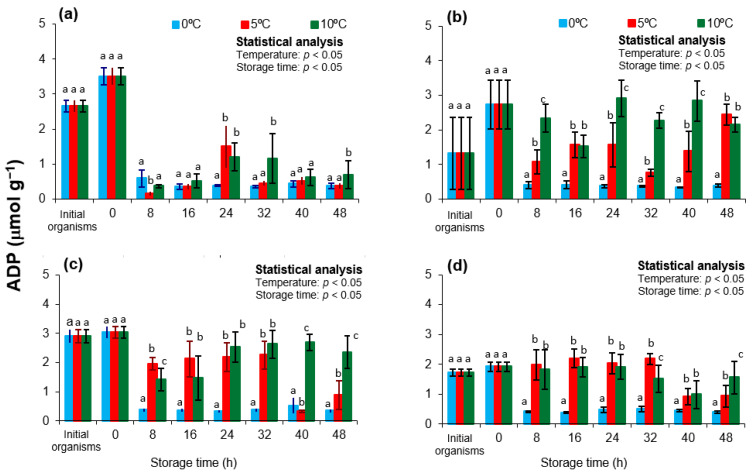
Adenosine diphosphate (ADP) concentrations in the adductor muscle of the lion’s paw scallop (*Nodipecten subnodosus*) harvested in different seasons and stored at three temperatures (0 °C, 5 °C, and 10 °C) for 48 h. Seasonal profiles are presented for (**a**) spring, (**b**) summer, (**c**) autumn, and (**d**) winter. Values are expressed as mean ± standard deviation (n = 6). “Initial” denotes specimens sampled directly from the lagoon at harvest, whereas “0 h” refers to samples collected immediately after transport to the shucking facility. Different lowercase letters above the bars indicate significant differences (*p* < 0.05) among storage temperatures at the same sampling time.

**Figure 3 animals-15-02953-f003:**
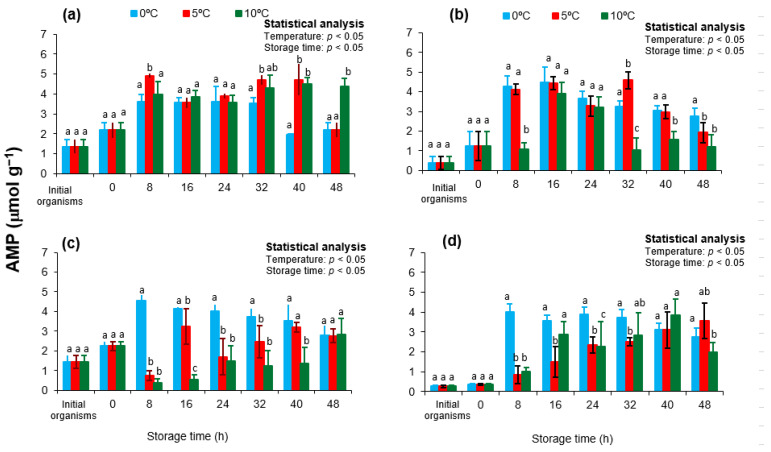
Adenosine monophosphate (AMP) concentrations in the adductor muscle of the lion’s paw scallop (*Nodipecten subnodosus*) harvested in different seasons and stored at three temperatures (0 °C, 5 °C, and 10 °C) for 48 h. Seasonal profiles are presented for (**a**) spring, (**b**) summer, (**c**) autumn, and (**d**) winter. Values are expressed as mean ± standard deviation (n = 6). “Initial” denotes specimens sampled directly from the lagoon at harvest, whereas “0 h” refers to samples collected immediately after transport to the shucking facility. Different lowercase letters above the bars indicate significant differences (*p* < 0.05) among storage temperatures at the same sampling time.

**Figure 4 animals-15-02953-f004:**
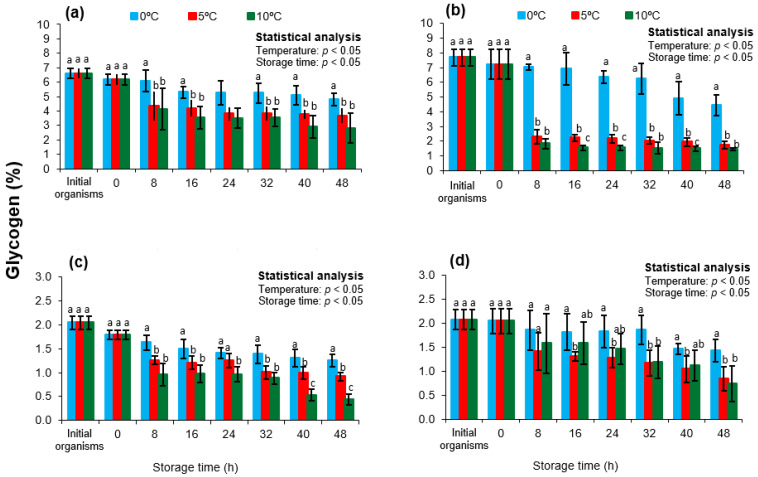
Glycogen concentrations in the adductor muscle of the lion’s paw scallop (*Nodipecten subnodosus*) harvested in different seasons and stored at three temperatures (0 °C, 5 °C, and 10 °C) for 48 h. Seasonal profiles are presented for (**a**) spring, (**b**) summer, (**c**) autumn, and (**d**) winter. Values are expressed as mean ± standard deviation (n = 6). “Initial” denotes specimens sampled directly from the lagoon at harvest, whereas “0 h” refers to samples collected immediately after transport to the shucking facility. Different lowercase letters above the bars indicate significant differences (*p* < 0.05) among storage temperatures at the same sampling time.

**Figure 5 animals-15-02953-f005:**
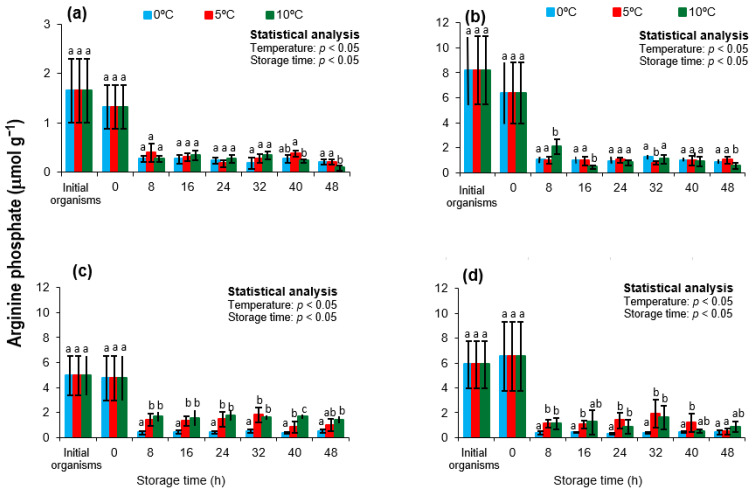
Arginine phosphate (Arg-P) concentrations in the adductor muscle of the lion’s paw scallop (*Nodipecten subnodosus*) harvested in different seasons and stored at three temperatures (0 °C, 5 °C, and 10 °C) for 48 h. Seasonal profiles are presented for (**a**) spring, (**b**) summer, (**c**) autumn, and (**d**) winter. Values are expressed as mean ± standard deviation (n = 6). “Initial” denotes specimens sampled directly from the lagoon at harvest, whereas “0 h” refers to samples collected immediately after transport to the shucking facility. Different lowercase letters above the bars indicate significant differences (*p* < 0.05) among storage temperatures at the same sampling time.

**Figure 6 animals-15-02953-f006:**
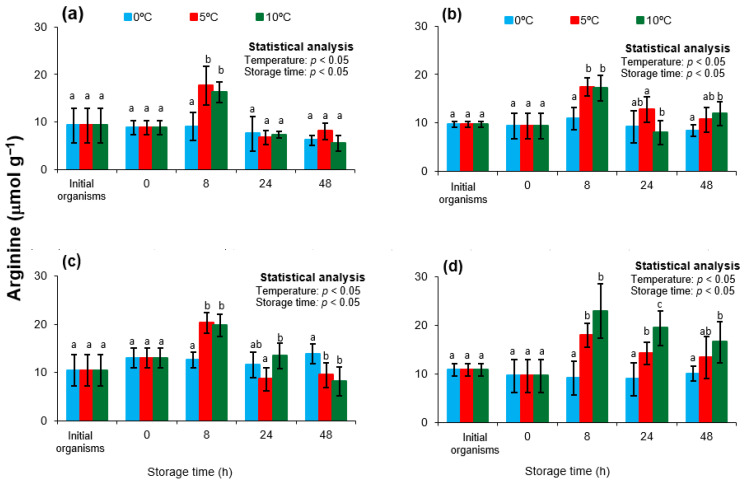
Free arginine in the adductor muscle of the lion’s paw scallop (*Nodipecten subnodosus*) harvested in different seasons and stored at three temperatures (0 °C, 5 °C, and 10 °C) for 48 h. Seasonal profiles are presented for (**a**) spring, (**b**) summer, (**c**) autumn, and (**d**) winter. Values are expressed as mean ± standard deviation (n = 6). “Initial” denotes specimens sampled directly from the lagoon at harvest, whereas “0 h” refers to samples collected immediately after transport to the shucking facility. Different lowercase letters above the bars indicate significant differences (*p* < 0.05) among storage temperatures at the same sampling time.

**Table 1 animals-15-02953-t001:** Adenylate energy charge (AEC) values in the adductor muscle of the lion’s paw scallop (*Nodipecten subnodosus*) harvested in different seasons and measured before and after transport to the shucking facility.

Stage	Spring	Summer	Autumn	Winter
Harvest	0.58 ± 0.02 ^a^	0.84 ± 0.11 ^be^	0.65 ± 0.02 ^c^	0.85 ± 0.02 ^b^
*Post* transport	0.31 ± 0.02 ^d^	0.67 ± 0.08 ^e^	0.51 ± 0.05 ^f^	0.83 ± 0.02 ^b^

Values are expressed as mean ± standard deviation (n = 6). Means within the same row or column followed by different superscript letters indicate statistically significant differences (*p* < 0.05).

## Data Availability

The data presented in this study are available on request from the corresponding author.

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
