# Peer review of "Energy Processes During *Rigor Mortis* in the Adductor Muscle of the Lion’s Paw Scallop (*Nodipecten subnodosus*): Effects of Seasonality and Storage Temperature"

_animals, 2025, doi:10.3390/ani15202953_

Round 1
Reviewer 1 Report
Comments and Suggestions for Authors
This manuscript presents a comprehensive and well-designed study on the postmortem energy metabolism of the commercially important lion’s paw scallop, Nodipecten subnodosus. The research addresses a relevant topic for the seafood industry, investigating the combined effects of seasonality and storage temperature on key biochemical markers related to rigor mortis and freshness. The experimental design is robust, with a clear factorial structure (4 seasons x 3 temperatures x multiple time points) and adequate replication. The findings are significant, as they challenge the common practice of immediate ice storage (0°C) by demonstrating that moderate refrigeration (5-10°C) better preserves energy status and delays rigor onset. The manuscript is generally well-written, and the data support the conclusions. However, therefore, There are still some issues that need to be refined as follows:
- The detection data of ATP, ADP, and AMP levels are critically important, as they directly impact the entire paper's results. Therefore, it is recommended that the article include the HPLC chromatograms of both the standard substances and the sample for ATP, ADP, and AMP.
- Clarity on "Transport" Effect:The impact of "transport" is a key factor, but the description of the transport conditions is vague ("ambient conditions," "damp cloth"). To better interpret the stress induced by transport, please provide more details: approximate duration of transport, range of "ambient" temperatures during transport for each season, and any humidity control measures. This context is important for understanding the "post-transport" (0h) data.
- The Results and Discussion section:
3.1 The current structure separates the "Results" (primarily descriptive narrative with some interpretation) from a very brief "Conclusions" section. A full Discussion section is missing. The interpretations woven into the "Results" should be moved to a dedicated Discussion section and expanded upon.
3.2 Compare and contrast the results with findings from other scallop and mollusk species cited in the introduction.
3.3 Provide more in-depth mechanistic explanations for the observed phenomena (e.g., why enzyme activity is better preserved at 5-10°C than at 0°C).
3.4 Discuss the practical implications of the study more deeply.
- Line 65: Change "aquaculture-based supply" to "aquaculture-based supply", the same as line 486.
- Line119,123: (0h) and 0℃ are modified to (0 h) and 0 ℃, the same as line 289.
- Line 196: (P < 0.05) is changed to (p < 0.05), the same below.
- Line 252,278: Please check the citation format of the references.
- It is suggested that a), b), c), and d) in Figure 4 be moved upwards a little.
Reviewer 2 Report
Comments and Suggestions for Authors
All comments are attached in the PDF file as sticky notes.

Round 2
Reviewer 2 Report
Comments and Suggestions for Authors
No further comments.